# Improving the Clinical Application of Natural Killer Cells by Modulating Signals Signal from Target Cells

**DOI:** 10.3390/ijms20143472

**Published:** 2019-07-15

**Authors:** Monika Holubova, Martin Leba, Hana Gmucova, Valentina S. Caputo, Pavel Jindra, Daniel Lysak

**Affiliations:** 1Biomedical Center, Faculty of Medicine in Pilsen, Charles University, Pilsen 323 00, Czech Republic; 2Faculty of Applied Science, University of West Bohemia, Pilsen 301 00, Czech Republic; 3Department of Haematology and Oncology, University Hospital Pilsen, Pilsen 304 60, Czech Republic; 4Centre for Haematology, Department of Medicine, Imperial College London, London, W12 0NN, UK

**Keywords:** NK cells, NKG2D ligands, HLA-KIR, relapsed AML

## Abstract

Relapsed acute myeloid leukemia (AML) is a significant post-transplant complication lacking standard treatment and associated with a poor prognosis. Cellular therapy, which is already widely used as a treatment for several hematological malignancies, could be a potential treatment alternative. Natural killer (NK) cells play an important role in relapse control but can be inhibited by the leukemia cells highly positive for HLA class I. In order to restore NK cell activity after their ex vivo activation, NK cells can be combined with conditioning target cells. In this study, we tested NK cell activity against KG1a (AML cell line) with and without two types of pretreatment—Ara-C treatment that induced NKG2D ligands (increased activating signal) and/or blocking of HLA–KIR (killer-immunoglobulin-like receptors) interaction (decreased inhibitory signal). Both treatments improved NK cell killing activity. Compared with target cell killing of NK cells alone (38%), co-culture with Ara-C treated KG1a target cells increased the killing to 80%. Anti-HLA blocking antibody treatment increased the proportion of dead KG1a cells to 53%. Interestingly, the use of the combination treatment improved the killing potential to led to the death of 85% of KG1a cells. The combination of Ara-C and ex vivo activation of NK cells has the potential to be a feasible approach to treat relapsed AML after hematopoietic stem cell transplantation.

## 1. Introduction

Acute myeloid leukemia (AML) is a severe hematological disorder that is mostly prevalent in the adult population. AML is characterized by expansion of cells arrested at different stages of differentiation of myeloid or/and monocyte lineage. The current standard treatment is chemotherapy, followed by allogeneic hematopoietic cell transplantation (allo-HSCT). AML relapse after allo-HSCT is still one of the most serious post-transplant complications with a very poor prognosis and without a clear treatment strategy. High-dose Ara-C (HiDAC) is mostly used for relapse treatment either alone or in combination with agents such as mitoxantrone, cladribine, and fludarabine [1]. This re-induction decreases a tumor burst but fails to maintain it long-term. Donor lymphocyte infusion or a second round of HSCT is a standard immunotherapy after re-induction to improve survival after HSCT; however, this has been without considerable success [2]. Modern cellular therapy has become a widely used treatment for many hematologic malignancies, especially in relapsing or refractory diseases. CD19 specific CAR T-cells have the potential to be a new standard treatment for lymphoid neoplasia, while CAR therapy in myeloid malignancies has the major limitation of the absence of specific targetable cell surface markers [3]. Infusions of ex vivo activated NK cells are promising treatment for myeloid leukemia, but the optimization of the medical products’ preparation and patient management are still missing.

NK cells as a part of the innate immune response can kill tumor or virus-infected cells presenting modified surface proteins. NK cells can recognize pathological cells by a wide spectrum of receptors responsible for positive or negative signals. These receptors can be divided into two main groups–activating (e.g., C-type lectin receptors—NKG2D/E/F, natural cytotoxicity receptors (NCRs)—NKp46, NKp30, and NKp44; activating KIRs—aKIRs) and inhibitory (inhibitory KIRs—iKIRs, C-type lectin receptors such as NKG2A/B) [4]. A fine balance between these receptors is critical for regulation of NK cell function. Inhibitory receptors can directly inhibit the activating pathway through the production of SHP-1 andSHP-2 phosphatases responsible of the dephosphorylation of the DAP10 or DAP12 axis or dephosphorylation of the Vav1 protein responsible for actin formation [5,6]. An increased activating signal that exceeds inhibitory regulation causes full activation of NK cells. The same effect can be obtained by a decrease of inhibitory signal by, for example, reducing HLA class I expression. Leukemia, lymphoma, and liver cancer express high levels of HLA class I, while most other tumors express low or no HLA class I [7]. Allogeneic stem cell transplantation using HLA–KIR mismatched donors is one approach to disrupt the NK inhibition through inhibitory KIRs while preserving their cytotoxic function. Unfortunately, the number of suitable donors is limited, and only a small proportion of patients can find an HLA-matched donor that is also KIR–HLA mismatched. Alternatively, a KIR blocking antibody is used to disrupt the binding of the HLA class I molecule and activation of the inhibitory pathway [8].

Tumor or virus-infected cells have different proteomic profiles compared with healthy cells, and these changes can be recognized by immune cells as signs of cell damage. The main proteins involved in the damage’s pathway are called damage-associated molecular patterns (DAMPs) and comprise many proteins overexpressed after chemical, physical, or biological stresses. DAMPs are mostly recognized by innate immune receptors, such as toll-like receptors but also by receptors on the surface of specific immune cells like CD40 on B cells (HSP70), CD91 on antigen-presenting cells (gp96, calreticulin), or one of the most potent activating receptor NKG2D on T/NK cells [9,10]. The ligands for NKG2D receptor are stress-inducible proteins involved in DNA damage response, which increase during viral infection, cancer or mitosis [11]. The ligands can be induced by cell cycle interfering chemotherapy (Ara-C or fludarabine), by heat, irradiation and by induction of biological processes like cell differentiation induced by differentiation-promoting drugs (e.g., all-transretinoic acid—ATRA) [12,13,14]. All these processes are accompanied by upregulation of DAMPs and make tumor cells more visible to the immune system [15]. Ara-C (1-β-D-arabinofuranosylcytosine) is a standard drug used to treat AML and works by blocking DNA polymerase and the cell cycle leading to cell death [16]. Leukemic stem cells (LSCs) are a small part of leukemia cells resistant to standard treatment and are co-responsible for relapse of AML. New approaches with the potential to kill this cell subset have been tested in vitro (e.g., antihistamines or cardiac glycosides), but clinical data are still missing [17]. A correlation between resistance and increased expression of NKG2D ligands has been shown [18]. The LSC compartment reacts to Ara-C treatment by upregulation of NKG2D ligands [19]. Cytostatic treated cells do not die but become sensitive to the immune response through the interaction of DAMPs and activation receptors on NK cells. Shifting the balance of NK cell signaling through boosting its activating component and blocking the inhibitory one is a way to induce an anti-leukemia response even in the presence of a strong inhibitory signal. The study aimed to test how to potentiate NK cell responses against AML cells in a clinical setting. 

## 2. Results

### 2.1. NK Cell Preparation

NK cells were isolated by immunomagnetic beads from a cohort of donors. The purity of the isolated NK cells was over 90% in all samples. After 10 days of culture with feeder cells, the median of NK cell purity was 82% (range 68–92%; Appendix A). All samples showed expression of activation markers CD25 or NKp44 (Figure 1A). The median of CD25 expression was 41% (range 33–73%) and of NKp44 was 36% (range 6–44%). Mutually exclusive expression of CD25 and NKp44 was observed (*p* ≤ 0.05). The expression of NKG2D ranged from low to high intensity, with a median mean fluorescence intensity (MFI) of 2498 (range 947–5168, Figure 1B). The presence of inhibitory KIR differed between donors. Only two of eight donors expressed inhibitory KIR2DL5 with no correlation on the effect of NK cells´ cytotoxic functions. All donors expressed KIR2DL1, KIR2DL2, KIR2DL3, KIR3DL1, and KIR3DL2 with high variability (2–40% of positive NK cells, Figure 2A,B). The correlation (Pearson’s *R*) between KIR levels and the number of dead KG1s cells did not show a significant association between KIRs expression and killing ability.

### 2.2. Change in the Expression of NKG2D Ligands after Ara-C Application

The expression of NKG2D ligands (ULBP1/2) in KG1a cells was measured by qPCR and normalized to untreated and *B2M* gene (ΔΔ*C*T). Time course experiment showed that after 24 h of Ara-C treatment (0.5µM), only *ULPB2* increased expression (ΔΔ*C*T = 1.14). At later time points (48 and 72 h), all other tested genes (*ULBP1-3, MICA/B*) showed increasing and maintained patterns of expression. At 48 h, both *ULPB1* and 2 increased their relative expression to almost 2 times (2.2 and 1.93, respectively). At this time point, *MICA/B* expression was also elevated (ΔΔ*C*T was 1.5 for *MICA* and 1.23 for *MICB*) after 0.5 µM Ara-C application. For all 5 genes tested, the highest induction was achieved at 72 h, where ΔΔ*C*T was 3.2 for both *ULBP1/2*, and *ULBP3* also reached higher levels (ΔΔ*C*T = 1.6). The kinetics of *MICA* and *MICB* induction was similar and reached to ΔΔ*C*T = 1.6 or 1.7 (all results are summarized in Figure 3).

An increase in mRNA levels of two selected ligands, MICA/B, with a low increase of mRNA levels correlated with increased cell surface expression using flow cytometry. After 24 h (the first time point when KG1a cells were added to the NK cell culture), there was no change of cell surface expression. A clear increment in cell surface expression was only detected after 48 h of treatment (Figure 4). The mean fluorescence intensity (MFI) was slightly higher in treated cells (MFI of positive population = 1796) compared with control (MFI of positive population = 1321) (see Appendix A).

### 2.3. Target Cell Preparation and Cytotoxic Potential of NK Cells

NK cell killing activity was estimated against treated and untreated KG1a cells. KG1a were pretreated with 0.5 µM Ara-C for 24 or 48 h respectively, and then co-cultured with NK cells for 8 or 24 h. HLA-ABC blocking antibody was always added 24 h before co-culture. The effect of blocking antibody was measured using flow cytometry and detection antibodies, which were prevented to bind the cells incubated for 24 h with a blocking antibody (Figure 5). The expression of HLA-ABC was evaluated during the co-culture experiment, where the epitope remained blocked during the entire experiment. KG1a cells were detected according to CD34 expression, and the number of dead cells was estimated as 7AAD positive cells from the entire CD34 positive population (Figure 6).

The viability of untreated KG1a cells was always higher than 98%. The presence of the anti-HLA class I antibody did not affect the viability of KG1a cells in culture without NK cells. Ara-C increased the number of dead KG1a in a time-dependent manner, and co-culture times were: T1 = 9.3%, T2 = 12.8%, T3 = 15.1%, T4 = 38% (Figure 7). The median of killing ability of NK cells against untreated cells was: T1 = 8.4%, T2 = 15.1%, T3 = 14.3%, T4 = 21.4%. The combination of chemotherapy and NK cells produced high numbers of dead KG1a cells with the maximum reached at the last time point (to 80%). In previous time-points, the numbers of dead cells were 65% for T3, 48% for T2, and 29% for T1. We did not find any correlation between the killing ability and NK cell activation of receptors´ levels.

The addition of a blocking antibody positively affected NK cell killing activity and further slightly improved the killing potential when combined with Ara-C (Figure 7). At the first time point, the percentage of dead cells after antibody treatment only was the same as after Ara-C (28.9%). Subsequent time-points showed lower potential of HLA blocking compared to Ara-C. The percentage of dead KG1a cells ware 27.3% for T2, 47% for T3, and 53.4% for T4. The combination of both treatments was the most efficient in all time points. Almost all the cells were killed at the last time point where the percentage of dead KG1a was 85%. In previous time points, the proportion of dead cells was as followed: T1 = 45.6%, T2 = 69.3%, T3 = 75.7%. All results are summarized in Appendix A and Figure 7A–D. We did not observe any correlation between inhibitory KIR expression and the killing ability. The expression of CD16 also did not influence the percentage of dead cells either (data not shown).

## 3. Discussion

NK cells are a crucial part of the anti-leukemia immune response after hematopoietic stem cell transplantation. The NK cell activity correlates with relapse-free survival in AML patients [20]. These data suggest that NK cells may play a crucial role in the control of leukemia development and relapse [21], therefore, donor NK cell infusion following HSCT might improve the outcome of patients. The ability of NK cells to kill residual or relapsed leukemia cells depends on the strength of activating and inhibitory signals. Ex vivo activation can induce expression of activating receptors, causing an exceeding signal from inhibitory receptors and full activation of their cytotoxic activity/potential [20]. Many protocols have been developed for preparing of NK cell-based medical products. However, optimal product characterization has not been defined yet. The key factors involved in NK cell therapy success are cell dosage and activation status [22]. We developed an ex vivo expansion protocol for preparing of NK cells, which was able to provide us with a sufficient number of NK cells with a high activation status. Using of cryopreserved mononuclear cells as an input material allows allowed us more flexible timing of NK cells application and treatment with multiple doses of fresh cells. NK cells are very sensitive to cryopreservation and could lose their recovery potential and activating state. Therefore, they still need the IL-2 re-activation [23]. Our in vitro activated NK cells isolated from cryopreserved mononuclear cells (MNCs) induced key activating receptors such as CD25, NKp44, or NKG2D. CD25 is mainly required for cell proliferation [24]. Our previous finding showed a reverse correlation between CD25 and NKp44 expression, where cells with high CD25 expression had low expression of NKp44 and vice versa [23]. CD25 is expressed mainly after the first days upon activation and is lost after 2 weeks of culture whereas NKp44 is stably expressed on the surface of over 50% of cells (data not shown). No correlation between cytotoxicity and expression of these markers was found. The activating receptor mentioned last—NKG2D—seems to be the most critical for NK cell response even when HLA class I molecule is present. AML is a heterogeneous disease with a highly variable expression of ligands for NKG2D on the cell surface [25]. The level of expression directly influences the clinical outcome of patients through NK cell activity to control relapse [26]. These ligands can be easily induced, pharmacologically leading to a better susceptibility of tumor cells to NK cells (examples of such treatment include HDAC-inhibitors or Ara-C) [19,27]. Ara-C is a cytostatic drug standardly used for the treatment of AML, but some AML cells could be resistant to this chemotherapy. This resistance is usually correlated with cellular stress responses that cause a higher expression of DAMPs, including NKG2D ligands [28]. We tested NK cell activity from healthy donors against the NK-resistant cell line, KG1a, untreated and treated with Ara-C (for induction of NKG2D ligands). Ara-C improved the killing ability in all our tested time point from about 17% to 58% when co-culture samples with treated and untreated target cells were compared. The strategy of chemotherapy as an inducer of DAMPs has been used in other cancers such as myeloma, low dose bortezomib [29]; prostate cancer, valproate [18]; or adenocarcinoma, Gefitinib [30]. Chemotherapy pretreatment is feasible within clinical protocols. In the case of AML, Ara-C chemotherapy used as standard (re)induction treatment could be followed by the application of ex vivo activated NK cells.

The HLA class I cells such as AML cells are resistant to NK cells’ killing ability because of a strong inhibitory signal and can escape NK cell responses [31]. HLA class I molecules react with inhibitory KIRs [32]. We tested several donors with a different expression of inhibitory KIR receptors and their cytotoxic potential against the NK-resistant leukemia cell line KG1a with a high HLA class I (ligands for KIR receptors) positivity. We did not find any correlation between the level of inhibitory KIRs and the cytotoxic potential of NK cells. However, inhibition of HLA–KIR interaction led to an increased NK cell killing potential of about 18% (time point 1) to 31% (time point 4). An improved killing activity of NK cells was observed in a previous study where pan-HLA blocking increased the proportion of dead cells by about 20% [33]. Another strategy to activate the killing potential of NK cells is to block inhibitory KIRs. The first clinical trials that blocked KIR–HLA interactions using lirilumab has already been reported, [34] with a promising result in preclinical testing [8]. However, this antibody binds only on KIR2DL1-3, and the KIR3DL1-3 inhibitory signal is still not blocked, so there is still the possibility of a high inhibitory signal for NK cells [35]. The expression of inhibitory KIRs is a dynamic process after hematopoietic stem cells transplantation, and it seems to be influenced by the presence of HLA ligands [36]. Therefore, the application of lirilumab can be limited by the presence and level of expression of inhibitory KIRs. There is no clinical-grade pan HLA-class I blocking antibody. Moreover, HLA-class I molecules are expressed in all nucleated cells, and their blocking could cause several side effects. In clinical practice, haploidentical NK cells with potential KIR–HLA mismatch are used to reduce KIR–HLA interactions [37,38]. 

Our study showed that blocking the inhibitory signal is not so efficient as the activating signal increases. The maximum proportion of dead cells in blocking experiments was 53% in comparison to 80% in experiments using chemotherapy. The combination of both treatments reached 85%, which represents only a minor improvement of killing ability. The chemotherapy pretreatment is not dependent on inhibitory KIR expression (either level type), which makes it donor-independent treatment.

Our findings proved that NK cells could kill leukemia cells with high expression of HLA class I molecules; however, they have to be combined with the sensitization of target cells. Pretreatment with cytostatic cells can be crucial for the immunotherapeutic protocols where ex vivo activated NK cells are used. Ara-C, standardly used for AML treatment could improve NK cells´ killing ability and the clinical outcome of patients.

## 4. Methods

### 4.1. NK Cell Preparation

Preparation of NK cell is based on the approved protocol for a clinical trial with an EudraCT number: 2018-001562-42. Peripheral blood mononuclear cells (PBMNCs) from 8 healthy donors were isolated by gradient centrifugation and Ficoll-Paque solution (GE Healthcare, UK) and cryopreserved at concentration 15 × 10 ^6^/mL in PBS with 10% albumin (Albunorm, Octapharma, Manchester, UK) and 10% Dimethyl Sulfoxide-DMSO (Cryosure, Wak-Chemie, Steinbach, Germany). NK cells were isolated immediately after thawing of PBMNCs with an NK cell isolation kit (Miltenyi Biotech, Teterow, Germany). Manufacturer´s instructions were followed. Pure cells were seeded at a concentration 1 × 10 ^6^ to SCGM medium containing 10% FBS (Gibco, Paisley, Scotland), IL-2 (1000 UI/mL; Proleukin, Nuremberg, Germany) and irradiated (25 Gy) PBMNCs from healthy donors (10 × 10 ^6^ of pooled MNC from 5 donors) as a feeder. Cells were cultured for 10 days, fresh IL-2 was added every 2–3 days. After this period NK, cells were split into 96-well plates at a concentration of 350 × 10 ^5^ in 300 µL of a medium containing IL-2. The purity, activation, and inhibitory receptors were evaluated (see section flow cytometry). The study was approved by the ethics committee (joint committee Faculty of Medicine in Pilsen and Faculty Hospital Pilsen) on 4 September 2014. Signed informed consent was obtained from all individual participants included in the study.

### 4.2. Target Cells Preparation

The NK-resistant cell line KG1a (Sigma Aldrich, Germany) was used as target cells. This cell line expresses a high level of HLA class I molecules which bind to inhibitory KIRs (for HLA typing see the datasheet of cells line). Cells were cultured for 7 d in Iscove’s Modified Dulbecco’s Medium (IMDM) medium (Gibco) with 10% FBS and antibiotics (100 units/mL of penicillin, 100 µg/mL of streptomycin, and 0.25 µg/mL Amphotericin B; Gibco). Then the cells were seeded at concentration 1 × 10 ^6^/mL. Ara-C (0.5µM, Cytosar, Pfizer, USA) was added for 24, 48, and 72 h. Anti HLA-class I antibody (clone W6/32, Santa Cruz Biotechnology, Inc, Dallas, USA) was added to block KIR–HLA interactions at a concentration 20 µg/mL 24 h before addition to the NK cell culture. 

Blocking efficiency was tested by flow cytometry analysis. The effect of Ara-C on NKG2D ligands was evaluated by qRT-PCR and a part of them also by flow cytometry. For details, see below (Section 4.3 and Section 4.4).

### 4.3. Flow Cytometry Detection of Surface Markers and Cytotoxicity

Purity of NK cells were determined immediately after NK cells were isolated with a combination of antibodies—CD45-BV510 (BD Bioscience, San Diego, CA, USA) and CD3FITC-CD16/56-PE (Exbio, Prague, Czech Republic). The expressions of surface activating as well as inhibitory receptors were measured on the day of cell splitting. For activating receptors, cells were stained with anti CD45-BV510, CD3-Pacific Blue (Beckman Coulter, Brea, CA, USA), CD56-APCCy7, CD25-PECy7, NKp44-APC, NKp46-PerCPCy5.5 (all BioLegend UK Ltd., London, UK), CD16-FITC (Exbio), and NKG2D-PE (eBioscience, San Diego, CA, USA). All the above-mentioned cell suspensions were incubated with antibodies for 15 min and then washed with PBS in 300 g/5 min. The cell pellet was resuspended in 300 µL PBS and immediately measured on a FACSCanto II flow cytometer (Becton Dickinson, Belgium).

Inhibitory receptors were determined using anti-KIRs antibody—KIR2DL1-PE, KIR2DL2/DL3-APC, KIR2DL3-FITC, KIR3DL1-FITC, KIR3DL1/DL2-PE, KIR2DL5-APC, CD56-PE-Vio770 (all Miltenyi), CD3-Pacific Blue, CD45-KromeOrange (Beckman Coulter), and CD16-PerCP (Exbio). Cells were incubated at 4 °C for 10 min. After incubation, all tubes were washed with PBS at 300 g/5 min. Cell pellets were resuspended at 300 µL of PBS and immediately measured on FACSCanto II. The entire gating strategy is shown in Appendix A.

KG1a was evaluated for the expression of HLA-class I and selected NKG2D ligands. The HLA class I expressions before and after blocking were estimated using anti-HLA-ABC—PE antibody (Biolegend). MICA/B expression on the surface of KG1a before and after Ara-C treatment was evaluated using anti-MICA/B-BV711 (BD Bioscience, USA). In co-culture experiments, the mix of 7-actinomycin (7AAD; Exbio) and CD34-PECy7 (for detection of KG1a cells; Beckman Coulter) as well as HLA-ABC-PE (Biolegend) were added to co-culture suspension. Cells (cell suspensions) were incubated with antibodies for 15 min and then washed with PBS in 300 g/5 min. The cell pellet was resuspended in 300 µL PBS and immediately measured on a FACSCanto II flow cytometer.

Analysis of cytometry data was performed using FlowJo software (Tristar, Ashland, OR, USA). The percentage of cells positive for KIRs, CD25, NKp44, 7AAD (dead cells), MICA/B, and HLA-ABC were determined. Median of fluorescence intensity (MFI) was used for the evaluation of NKG2D expression to compare individual donors.

A complete list of antibodies is summarized in Appendix A.

### 4.4. Evaluation of NKG2D Ligand Expression—qRT PCR

KG1a cells (culture techniques above) were treated with Ara-C (0.5 μM) for 24, 48, and 72 h. Total RNA was isolated using RNeasy Mini Kit (Qiagen, Hilden, Germany) according to the manufacturer’s protocol. Quality and quantity of the extracted RNA were evaluated using a Synergy HTX (BioTek, Winooski, VT, USA) and samples were diluted to a similar concentration. A QuantiTect Reverse Transcription Kit (Qiagen, Hilden, Germany) was used for reverse transcriptions following manufacturer instructions. The primers used for detection of *ULBP-1*, *ULBP-2*, *ULPB-3*, *MICA*, and *MICB* and for beta2-microglobulin (*B2M*) as a reference gene are in Table 1. All PCR reactions were performed using a QuantStudio5 Real-Time PCR System (ThermoFisher Scientific, Waltham, MA, USA). A QuantiTect SYBR Green PCR Kit (Qiagen, Hilden, Germany) as a master mix for all PCR reactions was used with modified manufacturer protocol (real-time PCR and two-step RT-PCR using Applied Biosystems cyclers and other cyclers). For each PCR run, a master mix was prepared on ice in duplicate with 1× final concentration 2× QuaniTect SYBR Green PCR Master Mix, 1 μM of each primer, and 2 μL of cDNA in a total volume 25 μL. The thermal cycling conditions included an initial denaturation step at 95 °C for 15 min, 50 cycles at 94 °C for 15 s, 56–59.5 °C (MICA: 56°C; MICB: 58 °C, ULBP1-3: 59.5 °C) for 30 s and 72 °C for 30 s followed by a dissociation curve. Appropriate amplicons were verified on a 2% agarose gel. Threshold cycle (*C*t) values were determined using QuantStudioTM Design & Analysis Software. Relative expression was calculated as ΔΔ*C*t = (*C*t (target, test) – *C*t (reference, test)) – (*C*t (target, calibrator) – *C*t (reference, calibrator)). Fold change in expression was calculated as a 2^-ΔΔCt^.

### 4.5. Co-Culture Cytotoxic Assay

NK cells after 10 days of culture were seeded into 96-well plates in SCGM with 5% FBS and IL-2 (1000 IU/mL). KG1a were treated with Ara-C (0.5 µM) for 24 or 48 h, and anti-HLA-ABC was added 24 h before co-culture to NK cells. The proportion of NK cells and KG1a cells was 1:10 in all experiments. To exclude an influence of IL-2, KG1s cells were transferred to SCGM medium with IL-2 without the addition of NK cells. Samples were collected at different time points, as described in Table 2. The number of dead cells was measured using flow cytometry and 7AAD. The specific killing activity was evaluated as the proportion of dead cells in the co-culture experiment minus spontaneous dying in a well without NK cell addition. The control co-culture well (KG1a without treatment) was determined as a natural killing activity without any sensitization. The control well (NK cell co-culture with untreated KG1a) and wells with treated cells were compared to evaluate the effect of treatment.

### 4.6. Data Evaluation

Data were evaluated using The MatLab software (The MathWorks, Inc., USA). Non-parametric Mann–Whitney *U* test was chosen to determine a statistical difference between groups at *p* < 0.05. Correlation of observed parameters was determined by an evaluation of the correlation coefficient (Pearson’s R). A coefficient greater than 0.8 and *p* < 0.05 was considered as significant.

## Figures and Tables

**Figure 1 ijms-20-03472-f001:**
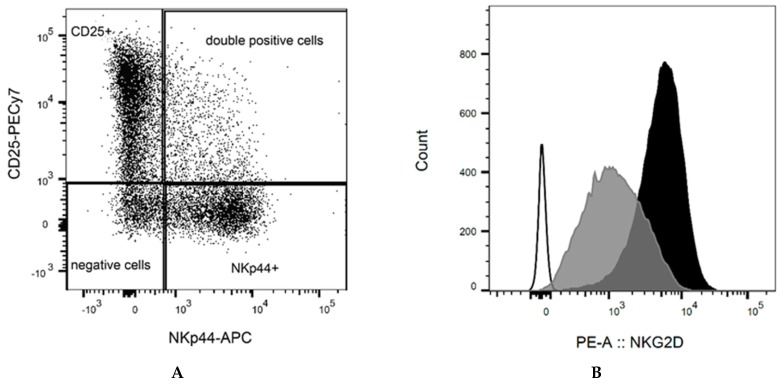
Representative flow cytometry plots of the activating receptor expressions on the surface of NK cells after 10 days of culture in the presence of IL-2 and pooled feeder cells. (**A**) Mutually exclusive expression of CD25 and NKp44. (**B**) Example of flow cytometry evaluation of NKG2D expression. All NK cells were positive with different fluorescence intensities. The grey peak represents low expression; the black peak corresponds to the donor with high expression. Unstained cells are displayed as the white peak.

**Figure 2 ijms-20-03472-f002:**
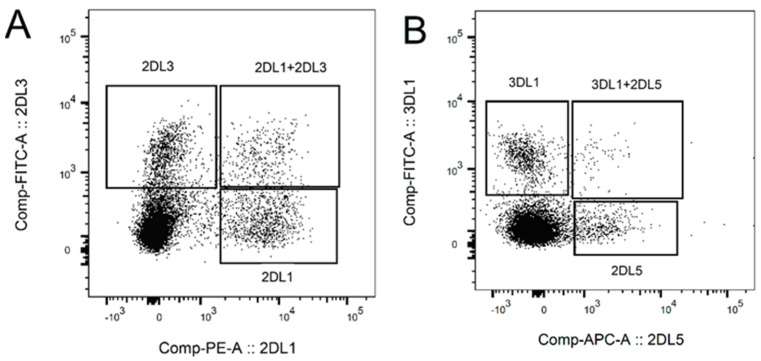
Characterization of surface expression of inhibitory KIR receptors. 2DL5 expression was detected only in two donors with a very low number of positive cells. The remaining KIRs expressed large variations. (**A**) A representative example of 2DL1 and 2DL3 expression; (**B**) Example of a positive sample for 2DL5 and 3DL1.

**Figure 3 ijms-20-03472-f003:**
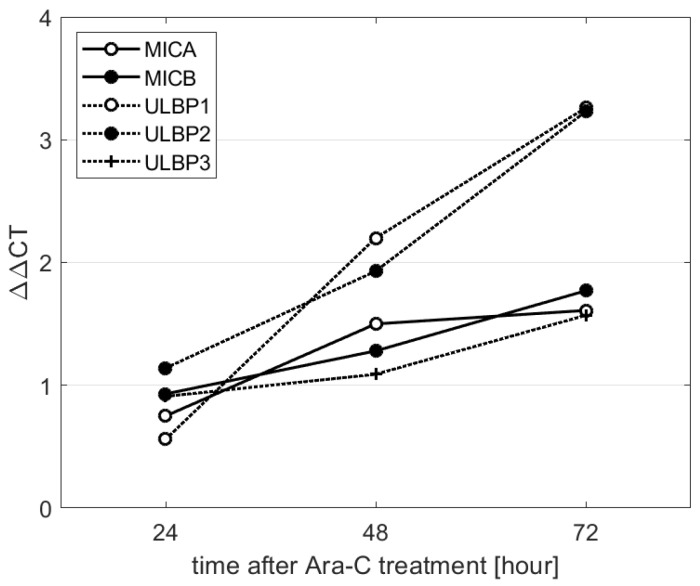
Dynamics of NKG2D ligand expression after treatment of KG1a with Ara-C (0.5 µM) for 24, 48, and 72 h. The effect of Ara-C was detected after 48 h in all genes except *ULBP3*. In the last time-point (72 h), the expression of all tested genes was higher.

**Figure 4 ijms-20-03472-f004:**
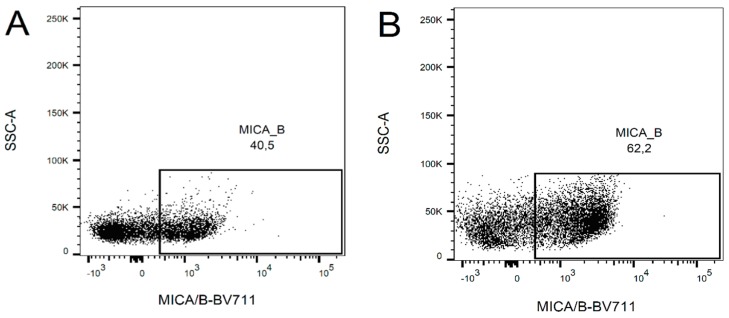
Example of surface expression of MICA/B in KG1a untreated (**A**) or treated (**B**) with Ara-C (0.5 µM) for 48 h. The treatment has induced an increase of expression of about 20% of positive cells.

**Figure 5 ijms-20-03472-f005:**
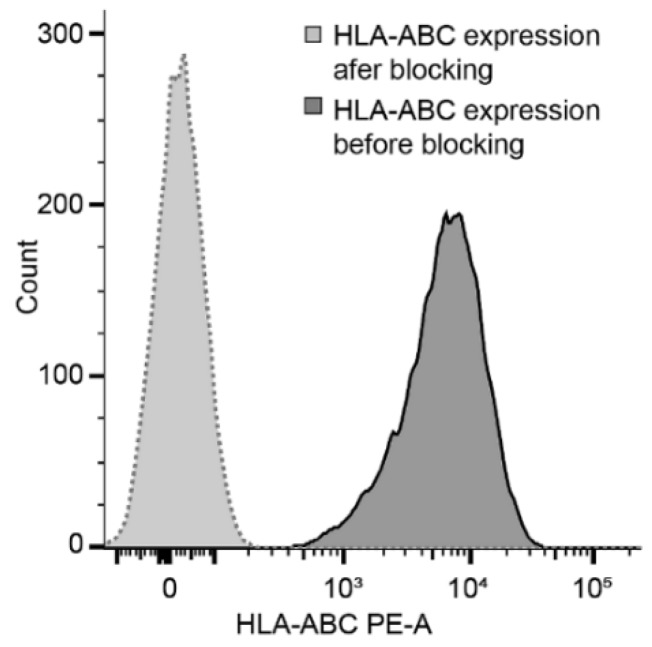
HLA-ABC expression on the surface of KG1a. The HLA class I molecules were blocked with anti-HLA class I blocking antibody for 24 h followed by washing and staining as previously described.

**Figure 6 ijms-20-03472-f006:**
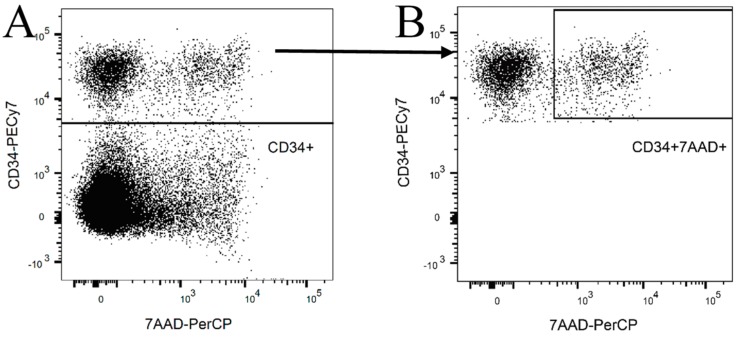
KG1 cell killing assay. Determination of dead KG1a cells co-cultured with NK cells. KG1a cells were selected based on their expression of CD34 (**A**) followed by gating of 7AAD from CD34-positive cells (**B**). The proportion of dead CD34 cells was compared between control co-culture and treated cells.

**Figure 7 ijms-20-03472-f007:**
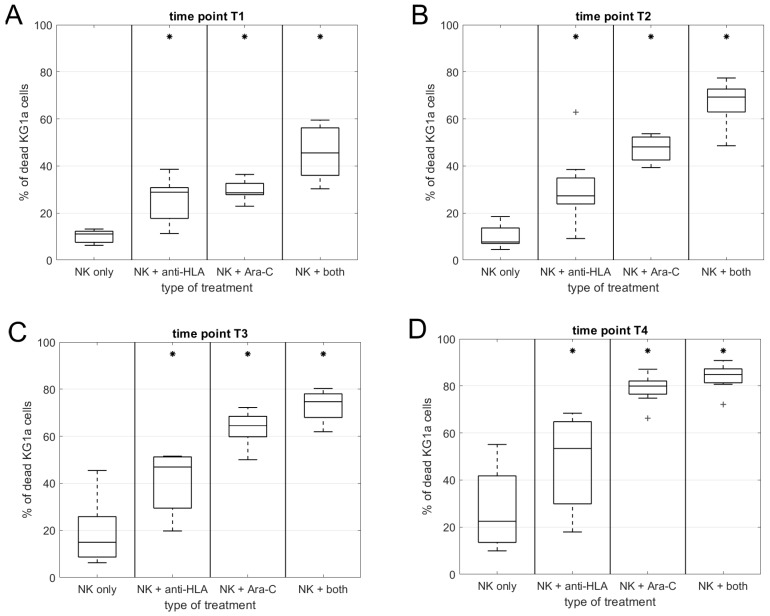
Evaluation of dead KG1a cells in culture with NK cells in the different culture conditions. NK cell killing against untreated cells was under 40% in almost all samples. Anti-HLA blocking increased the cytotoxic potential but was not comparable with Ara-C pretreatment, where 80% of KG1a cells were killed. Combination of both treatments showed the highest number of dead KG1a (85%), *n* = 8. Graphs (**A**)–(**D**) represent individual time points. (**A**) 24 h after Ara-C and 8 h of co-culture; (**B**) 24 h after Ara-C and 24 h of co-culture; (**C)** 48 h after Ara-C and 8 h of co-culture; (**D**) 48 h after Ara-C and 24 h of co-culture. For more details about time-points, see methods–in Section 4.5. *, *p* < 0.05; +, outliers.

**Table 1 ijms-20-03472-t001:** Primers used for the detection of NKG2D ligands and reference gene expression.

Assay	Name	Sequence 5′→3′
MICA	*MICA-F*	CTTGGCCATGAACGTCAGG
	*MICA-R*	CCTCTGAGGCCTCGCTGCG
MICB	*MICB-F*	ACCTTGGCTATGAACGTCACA
	*MICB-R*	CCCTCTGAGACCTCGCTGCA
ULBP1	*ULBP1-F*	GTACTGGGAACAAATGCTGGAT
	*ULBP1-R*	AACTCTCCTCATCTGCCAGCT
ULBP2	*ULBP2-F*	TTACTTCTCAATGGGAGACTGT
	*ULBP2-R*	TGTGCCTGAGGACATGGCGA
ULBP3	*ULBP3-F*	CTGATGCACAGGAAGAAGAG
	*ULBP3-R*	TATGGCTTTGGGTTGAGCTAAG
B2M	*B2M-F*	CTATCCAGCGTACTCCAAAG
	*B2M-R*	GAAAGACCAGTCCTTGCTGA

**Table 2 ijms-20-03472-t002:** Sample collections in co-culture experiments.

	Ara-C before Addition to NK Cells	HLA-Blocking before Addition to NK Cells	Co-Culture with NK Cells
T1	24 h	24 h	8 h
T2	24 h	24 h	24 h
T3	48 h	24 h	8 h
T4	48 h	24 h	24 h

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
