# Peer review of "Improving the Clinical Application of Natural Killer Cells by Modulating Signals Signal from Target Cells"

_ijms, 2019, doi:10.3390/ijms20143472_

Reviewer 1 Report

Presented study entitled as "Improving of Clinical Application of NK Cells by 3 Modulation of Signal from Target Cells", submited by Holubova et al.,  is quite simple yet interesting and elegant project and would be of interest in the field.Well designed and written, it deals with an important problems of immunotherapies. Studies of that type are appreciated because they can move the field forward.  Imunotherapies are considered to be a candidates for successfull future cancer therapy and articles like this one can make an impact to that fact.

Small English correction and editing  is advised  before publication.

Question for authors:

Questions 1: Did authors of proposed study use the isotype controls for detection antibodies?

Queastion 2: Better description of antibodies used in study will be appreciated - (in form of table).

Question 3: Did authors use blocking step in their phenotypisation experiments? If yes, add description, if not, explain.

Question 4:  Is 8 collected samples  sufficient for such type of study?

Overall, I recomend to accept proposed work for publication in IJMS journal.

Author Response

Dear reviewer,

many thanks for the questions and suggestions, they will surely improve the manuscript. We addressed your questions/comments point by point and made the corresponding changes were required. Please, see the attachment.

Regards,

Monika Holubova

(Corresponding author)

Response to Reviewer 1 Comments

Presented study entitled as "Improving of Clinical Application of NK Cells by 3 Modulation of Signal from Target Cells", submited by Holubova et al.,  is quite simple yet interesting and elegant project and would be of interest in the field.Well designed and written, it deals with an important problems of immunotherapies. Studies of that type are appreciated because they can move the field forward.  Imunotherapies are considered to be a candidates for successfull future cancer therapy and articles like this one can make an impact to that fact.

Point 1: Small English correction and editing is advised before publication.

English will be revised before sending the revise manuscript.

Question for authors:

Question 1: Did authors of proposed study use the isotype controls for detection antibodies?

Response 1: Isotype control were used just in case of activation receptors (NKG2D, NKp44, CD25) and MICA/B. The rest of gating was based on positive/negative strategy. Staining with isotype control is showed in supplementary fig.1A-F.  We updated the methods for flow cytometry and added whole gating strategy to supplementary methods.

Question 2: Better description of antibodies used in study will be appreciated - (in form of table).

Response 2: Antibodies in details are shown in supplementary table 1.

Question 3: Did authors use blocking step in their phenotypisation experiments? If yes, add description, if not, explain

Response 3: All cells were stained after splitting when the pellet was resuspended in culture medium containing FBS. Therefore, there is no need of additional blocking.

Question 4: Is 8 collected samples sufficient for such type of study?

Response 4: The study was performed sequentially, and data were evaluated immediately. Since all samples behaved in the same way and their number was sufficient for statistical analysis, we decided to terminate the study and evaluate it overall. The results of the study are now being used in the clinical protocol.

Reviewer 2 Report

The manuscript describes improved NK cell killing activity encountering target cells treated with Ara-C in combination with an anti-HLA blocking antibody. Based on these observations, the authors suggest that the combination of this drug and ex vivo activation of NK cells could be a useful approach to treat relapsed AML.

The manuscript lacks originality. The capability of chemotherapy drugs, including Ara-C to induce NKG2D ligands on cancer cell lines, such as KG-1, as well as the effect of anti-hla antibodies to mask major histocompatibility complex gene products on tumor cells to enhance susceptibility of these cells to lysis by natural killer cells have been already described.

The following issues should be improved by the authors:

-Explain the choice of the protocol of NK cell isolation and expansion

-Fig. 1 is just about purity of NK cell preparation, so it could be included in supplementary materials.

-Fig. 4 is just about the activity of anti-hla antibodies, it could be included as a control in Fig.8.

- Fig. 6 shows the percentage of untreated and Ara-C treated KG1a cells expressing MICA/B. Why didn’t you use specific antibodies for MICA and MICB? Could you show data as an

Histogram with mean? Could you show also data facs about ULBPs?

-Fig. 7 is just a gating strategy so it could be shifted in supplementary materials.

-Fig. 9 is simply a different representation of Fig. 8. Data could be shown just by using a figure.

- Fig. 8, blocking anti NKG2D antibodies should be included to demonstrate that increased susceptibility of Ara-C KG1a cells to NK cells depends on NKG2D ligand upregulation.

-Data need to be improved by extending observations to primary AML cells.

Author Response

Dear reviewer,

many thanks for the questions and suggestions, they will surely improve the manuscript. We addressed your questions/comments point by point and made the corresponding changes were required. Please, see the attachment.

Regards,

Monika Holubova

(Corresponding author)

Response to Reviewer 2 Comments

The manuscript describes improved NK cell killing activity encountering target cells treated with Ara-C in combination with an anti-HLA blocking antibody. Based on these observations, the authors suggest that the combination of this drug and ex vivo activation of NK cells could be a useful approach to treat relapsed AML.

The manuscript lacks originality. The capability of chemotherapy drugs, including Ara-C to induce NKG2D ligands on cancer cell lines, such as KG-1, as well as the effect of anti-hla antibodies to mask major histocompatibility complex gene products on tumor cells to enhance susceptibility of these cells to lysis by natural killer cells have been already described.

The goal of our work is to show a direct comparison of two different approaches of target cells modulation and their combination for much needed translational proposes. Beside this comparison, we present here complete study consist of the protocol for NK cells preparation for clinical use as well as for conditioning of patients. Today´s NK cells applications are mostly based on KIR-HLA mismatch (by using haploidentical NK cells), but we showed here the importance of increased activation signal instead of decreased an inhibitory one used in clinical trials. Therefore, our study is original in the way of complete information appreciated in clinical field and we believe that this study can move the field forward.

The following issues should be improved by the authors:

Point 1: Explain the choice of the protocol of NK cell isolation and expansion

Response 1: The protocol for culture of NK cells is based on a series of preliminary experiments comparing activation of NK cells by IL-2 only, by IL-2 and autologous feeder cells and by IL-2 and allogeneic feeder cells (single batch or pooled). The final protocol is already used in ongoing clinical trial where NK cells are prepared for the prevention and treatment of relapsed AML (EudraCT number: 2018-001562-42). We have included this clarification in the discussion.

Point 2: Fig. 1 is just about purity of NK cell preparation, so it could be included in supplementary materials.

Response 2: We added whole gating strategy to supplementary methods and transfer the figure to supplementary results.

Point 3: Fig. 4 is just about the activity of anti-hla antibodies, it could be included as a control in Fig.8

Response 3: This Figure is important because it shows the capacity of the selected antibody for blocking experiments to completely cover the HLA-ABC cell surface molecules. We transferred this figure (in revised manuscript Figure 5) to renamed part 2.3. Target cells preparation and cytotoxic potential of NK cells

Point 4: Fig. 6 shows the percentage of untreated and Ara-C treated KG1a cells expressing MICA/B. Why didn’t you use specific antibodies for MICA and MICB?

Could you show also data facs about ULBPs?

Response 4: Ara-C is known to induce NKG2D ligands expression. In figure (in revised manuscript Figure 3), we show the time course and comparison of five target genes by qPCR, a very sensitive method.

As a validation, in Figure 4, we show that 2 of the lowest responding genes (MICA/B), show clear increased protein membrane expression at an intermediate time point (48 hrs).

Point 5: Could you show data as an histogram with mean?

Response 5: We add histogram where you can see fluorescence intensity which is slightly changed after treatment in positive cells whose presence increased after Ara-C application. We add MFI (mean fluorescence intensity) evaluation to revised manuscript. See supplementary figure 3.

Point 6: Fig. 7 is just a gating strategy so it could be shifted in supplementary materials.  

Response 6: We prefer to leave this figure (in revised manuscript Figure 6) in main manuscript to show very clearly how the number of dead cells was evaluated. Based on our experience, it is important for better understanding of results.

Point 7: Fig. 9 is simply a different representation of Fig. 8. Data could be shown just by using a figure.

Response 7: The figure 8 shows the results in time dependent manner for clear comparison between treatments and time points, while figure 9 shows boxplots with better description of the dataset at each time point. For clarity, we transferred figure 8 to supplementary results.

Point 8: Fig. 8, blocking anti NKG2D antibodies should be included to demonstrate that increased susceptibility of Ara-C KG1a cells to NK cells depends on NKG2D ligand upregulation.

Response 8: The suggested experiment could support our theory, but it needs new series of experiments that we are unable to do in 10 days deadline for re-submission. Moreover, we claim that Ara-C sensitizes target cells to NK, but we did not at this point claimed that it is directly dependent on NKG2D and its ligands. We hypothesize about this based on our data and relative literature. We tested if there is some correlation of NKG2D expression and killing ability of NK cells and we found no correlation. But still, we found increased expression of NKG2D ligands after Ara-C and the treated cells showed higher susceptibility to NK cells, therefore we could suggest that there could be connection. We ensured that in the manuscript there is no mention of direct dependency on NKG2D expression, but we are saying the increased expression of DAMPs is improving target cells susceptibility.

Point 9: Data need to be improved by extending observations to primary AML cells

Response 9: This is a very important point, and we are certainly aware of its value. We already tried to test our protocol for primary AML cells, but primary AML cells are hard to culture, and they started to die within few days (3-5days) spontaneously (or when we cryopreserved them within 1-2 days after thawing) with downregulation of CD34 or CD117 (possible detection markers). Because NK cells have to be cultured for 10 days, it has been impossible to co-culture them in almost all cases. In two cases, where NK cells and primary AML cells (treated or untreated) we culture together, we saw a hint of the same trend like in experiments with KG1a but because of a very low quality and a credibility of these measurements, these data cannot be published. In ongoing clinical trial, we planned to do this and will collect fresh AML cells at different stages. We would like to test fresh AML cells (in patients with hematological relapse before re-induction) in co-culture with NK cells that will already be prepared to treat the patient (NK cells are being prepared in a very early phase of relapse). These experiments are planned within the currently launched clinical trial.

 Round  2

Reviewer 2 Report

The revised version of the article can be accepted in present form  

Author Response

Thank you very much. We are pleased that the reviewers were satisfied with our answers and that the revised manuscript is interesting for them and suitable for publication